# Experimental Models in Neovascular Age Related Macular Degeneration

**DOI:** 10.3390/ijms21134627

**Published:** 2020-06-29

**Authors:** Olivia Rastoin, Gilles Pagès, Maeva Dufies

**Affiliations:** 1Institute for Research on Cancer and Aging of Nice, CNRS UMR 7284, INSERM U1081, Centre Antoine Lacassagne, University Cote d’Azur (UCA), 06000 Nice, France; olivia.rastoin@unice.fr (O.R.); gilles.pages@unice.fr (G.P.); 2Biomedical Department, Centre Scientifique de Monaco, 98000 Monaco, Monaco

**Keywords:** neovascular age related macular degeneration (vAMD), wet AMD, in vitro model of AMD, mice and zebrafish model of AMD

## Abstract

Neovascular age-related macular degeneration (vAMD), characterized by the neo-vascularization of the retro-foveolar choroid, leads to blindness within few years. This disease depends on angiogenesis mediated by the vascular endothelial growth factor A (VEGF) and to inflammation. The only available treatments consist of monthly intravitreal injections of antibodies directed against VEGF or VEGF/VEGFB/PlGF decoy receptors. Despite their relative efficacy, these drugs only delay progression to blindness and 30% of the patients are insensitive to these treatments. Hence, new therapeutic strategies are urgently needed. Experimental models of vAMD are essential to screen different innovative therapeutics. The currently used in vitro and in vivo models in ophthalmic translational research and their relevance are discussed in this review.

## 1. Introduction

Age related macular degeneration (AMD) is a disease affecting the macular (central) region of the retina that occurs most frequently in elderly individuals. It is primarily caused by the degeneration of the retinal pigment epithelium (RPE). This region of the blood-ocular barrier transports nutrients and oxygen between the retina and the choroid and produces growth factors and cytokines. The RPE layer also plays a crucial role in phagocytosis of used photoreceptors. It forms a monolayer on top of Bruch’s membrane (a multi-layered extracellular matrix), under which are situated chorio-capillaries. The RPE and Bruch’s membrane act as a barrier between the choroid and the retina (Figure 1).

Two main AMD subtypes exist: dry AMD which is most prevalent and neovascular AMD (or wet AMD (vAMD)) which occurs in only 10–15% of cases. Though dry AMD is most common, vAMD is responsible for most of the loss of vision in patients [1].

The first signs of dry AMD are small, white-yellow deposits called drusen situated between the RPE and the underlying Bruch’s membrane. They are composed of an accumulation of ubiquitin, integrins, complement, collagen, fibronectins and beta-amyloids. Drusen can be classified by size (small: less than 63 µm in diameter, large: over 125 µm in diameter) and form (hard: well defined margins, soft: diffuse margins). The formation and accumulation of drusen cause hypoxia in the retinal pigment epithelium by decreasing the flow of oxygen from the chorio-capillaries to the RPE, which secondarily leads to degeneration. Drusen are typically not seen in murine AMD; however, more and more human-like models are starting to emerge [2].

In its late stages dry AMD is also characterized by a decrease in autophagy and clearance of debris by RPE cells [3]. Hypo/hyper pigmentation, or general displacement of pigmentation, is also observed in RPEs. The term ”geographic atrophy” is used when a loss of RPE and photoreceptor cells is observed in the central region of the retina, namely the macula. It is a hallmark of late-stage dry AMD, and the atrophic area progresses slowly over the years, decreasing visual function. The causes for geographic atrophy are yet still unclear, though environmental and genetic factors (specifically the complement pathway) play a substantial role [4].

vAMD occurs in 10 to 15% of patients and is characterized by choroidal neovascularization. Capillaries grow from the choroidal capillaries through Bruch’s membrane and into the retina [5]. The choroidal neovascularization (CNV) caused during vAMD leads to edema in the retina and to photoreceptor cell damage due to vascular leakage. Neovascularization causes RPE detachments, tears and hemorrhages and leads to atrophic macular scars that cause permanent damage to vision. In some serious cases it also causes the detachment of retinal pigment and accumulation of serous fluids and hemorrhage under the RPE layer. vAMD may in some cases also present drusen [1]. 

Most genetic variants known to cause heritable AMD are genes in the complement cascade or genes encoding for VEGF and components of the collagen matrix pathway [6]. For example, polymorphisms in complement factor H, I, and complement 3 and 9 genes are associated with AMD [7]. A coding variant was described in the factor I region (*CFI* gene) as highly penetrant (individuals carrying that variant also expressed the associated phenotype). FI is a serine protease, part of the complement cascade, that cleaves the alpha chains of activated complement factors C3b and C4b. It is constitutively expressed by RPE cells. The FI-mediated regulatory mechanisms are also associated with CFI variants determinant in atypical hemolytic uremic syndrome and systemic lupus erythematosus. The pGly188Ala substitution reduces the expression and secretion of FI which leads to loss of function mechanisms. Currently, new treatments selectively inhibiting complement activation are being developed [8].

## 2. Treatment of vAMD

vAMD is more aggressive than the dry form. Currently, the only treatment relies on intravitreal injections of anti-VEGF to inhibit angiogenesis and minimize visual loss. 

Three main drugs are currently approved: aflibercept (Eylea, a fusion protein of the extracellular domains of VEGFR1-2 serving as decoy receptors for VEGF, VEGFB and PlGF), bevacizumab (Avastin, a recombinant humanized monoclonal antibody that inhibits VEGF) and ranibizumab (Lucentis, a humanized monoclonal antibody fragment derived from the same parent antibody than bevacizumab). Both ranibizumab and bevacizumab have similar efficacy in randomized trials over a period of 24 months. However, bevacizumab is not approved for ophthalmic use in every country [9,10]. 

Injections are performed monthly for three months, and then every two months. Once there is a reduction in symptoms, each patient is followed individually in order to reduce the number of injections needed and to treat on a case to case basis. In a “treat-and-extend” regimen, monitoring visits are performed to fine tune the treatment and injections are performed during the visits. This method reduces the number of visits and the number of injections, reducing costs and the strain on patients which are often reluctant to undergo multiple injections. 

However, frequent intravitreal injections of anti-VEGF drugs are associated with ocular hypertension, retinal detachment, ocular infection, and poor patient compliance. Moreover, repeated injections can sometimes lead to intra ocular inflammation, infectious endophthalmitis, or RPE tearing [9,11]. 

A novel inhibitor called conbercept (Lumitin) developed by the Chengdu Kang Hong Biotech is currently in phase 3 clinical trials. Conbercept binds to several VEGF family members including VEGF-A, VEGF-B, and PlGF. It has a longer half-life and a better bioavailability than ranibizumab or aflibercept. It only needs quarterly administrations, thus reducing the load on patients and the healthcare system [12]. 

Other means of treating vAMD were used before the widespread of anti-VEGF injections. However, these methods were less reliable with relapses in following years and did not improve visual recovery. They include laser photocoagulation, which consists of laser treatment in the extra-foveal, juxta-foveal, or sub-foveal zone; and photodynamic therapy, which consists of a non-thermal laser treatment. A photosensitizer is injected into the area of interest and activated by a specific light wave. This method prevents the occurrence of thermal tissue damage [13]. 

Finally, vitamin supplements in the intermediate stages of the disease may delay the development of AMD in the other eye and the reduction of vision loss; specifically vitamins E, C, carotenoids, and mineral supplementation (zinc oxide and cupric oxide) [13].

The procedure of anti-VEGF treatment is still quite time consuming and traumatic for the patients. In addition, even if anti-VEGF treatments delay vision loss caused by vAMD, many patients nevertheless remain refractory to these treatments or become resistant. It is therefore absolutely necessary to develop new therapeutic strategies to treat patients in the first and second line. Thus, it is very important to implement different models of vAMD in order to reach this objective.

## 3. In Vitro Models of vAMD 

RPE cells are those affected by the vascular leakage, though there have been some models successfully replicating the choroid cells [14,15]. RPE cells are derived from human embryonic stem cells (hESC-RPE) or from the immortalized cell line ARPE-19. 

### 3.1. Human Primary RPE Cells

Primary RPE cells are fairly easy to isolate from either human cadaverous eye cups, or, more commonly, human fetal eyes (hfRPE). hfRPE cells present a similar mRNA expression profile to native RPE cells, in addition to features essential for RPE function (tight junctions, expression of specialized proteins, phagocytosis, and the ability to secrete multiple factors). The cells are dissociated from fetal eye cups that are sampled from fetuses around 10–22 weeks of gestation, and then trypsinized and seeded in vitro. Once differentiated, hfRPE cells seldom express variability despite different donors, making them a good in vitro model. However, hfRPE cells have a limited capability to divide and will eventually stop proliferating [16].

hfRPE cells can be cultivated without serum by supplementing the medium with B-27 (specific supplement). This method limits bias of undefined factors in the serum that can vary both in concentrations and between preparations. For clinical use, the exposure to animal products before a transplant is not recommended. A specific culture medium enables the accurate measurement of secreted factors and provides a controlled and reproducible environment. Cells cultivated in this medium displayed the same protein and gene expression profile as cells cultivated in the presence of serum [17].

Postnatal RPE cells, or cells obtained from cadaverous eye cups represent another source of primary cells. Cells from an adult eye already display a mature phenotype, despite having limited expansion potential compared to hfRPEs. The cells are dissected from the human eye in layers to preserve their junctional bonds. Human primary RPE cells have some disadvantages compared to ARPE-19 or iPSC-RPE cells described below. Human primary RPE cells undergo an epithelial to a mesenchymal transition over time preventing long-term culture. Moreover, their supply is limited for evident ethical concerns [18].

### 3.2. ARPE 19 Cells

ARPE-19 is a human RPE-cell line obtained from a 19-year old male donor. The cell line was spontaneously immortalized and has a rapid growth rate. Cells form polarized monolayers and display similar physiological properties as found in RPE cells in vivo; they exhibit pigmentation and have a cobblestone like appearance. 

They are also genetically similar to primary RPE cells: they are diploid, and they express specific proteins such as cellular retinaldehyde binding protein 1 (CRALBP) and the RPE-specific protein RPE65. Thus, ARPE-19 cells share a similar profile with primary RPE cells and are widely used in vitro. Despite these similarities, ARPE-19 cells may lose their specialized properties after multiple passages, notably if cultured in an unsuitable medium. Furthermore, cells have to be cultivated for over 4 months to obtain the closest similarities to native RPEs. Specifically, higher levels of P and E-cadherin, and downregulation of TGF-β, were found in cells cultured in these experimental conditions. The expression of *RPE65* (retinoid isomerohydrolase) and *CRALBP* genes was also highly increased. The authors suggest using low passage, long-term cultured ARPE-19 cells as a culture model [19].

When grown post confluency, APRE-19 cells create polygonal arrays of cells with a “cobblestone-like” appearance and display increased pigmentation and partial polarization (apical microvilli, junctional complexes, basolateral infoldings resembling microvilli, and a polarized distribution of many organelles)[20].

### 3.3. Stem Cell Derived RPE Cells (iPSC-RPE)

Stem cell derived RPE cells or induced pluripotent stem cells have the advantage of sharing a similar protein profile to ARPE-19 cells, and they closely model the function and metabolic activity of native RPE. They have the same genetic background as mature human RPE cells and exhibit the same morphological properties such as polygonal and pigmented morphology, polarity of protein expression and secretion, phagocytosis of photoreceptor outer segments, and maintenance of RPE phenotypes after transplantation into mouse retina. A higher number of cells can be obtained without invasive methods needed to derive RPE human cells from patients’ samples [21]. 

A small percentage of human embryonic pluripotent stem cells spontaneously differentiates into RPE cells. However, this method of culture has a poor yield and is time consuming, as it requires 2–3 months of growth until the pigmented RPE foci can be selected an expanded. Most RPE cells are currently derived from iPSC cultured in the presence of specific growth factors mimicking in vivo cues, thus reducing the growth time from several months down to around a fortnight. Some of these factors include basic fibroblast gGrowth factor (FGF), nicotinamide, activin A, IGF1, and VIP (vasoactive intestinal peptide) [22]. However, the use of growth factors in RPE differentiation is suboptimal for clinical uses.

Recently, Maruotti et al., 2015 [21] developed a protocol to differentiate iPSC-RPE cells that requires only two compounds to initiate differentiation. Nicotinamide, which was previously in use, and a compound called chetomin (CTM), a molecule that was identified by qPCR high-throughput screening from a high number of molecules that promote RPE differentiation, notably transcription factors MITF and OTX2. This led to a high yield of RPE cells after one month of differentiation. RPE-committed cells did not display any pigmentation or morphological characteristics. A simple medium change was sufficient to induce the apparition of the visual hallmarks of RPE cell morphology. They tested the clinical relevance of these cells by injecting them into the eyes of albino mice and observed that they were functional and did not exhibit tumor cell characteristics.

This protocol is currently widely used as it is time and cost efficient and does need growth factors.

iPSC-RPE cells share a similar background to human RPE cells, and thus can be used as a predictive model to characterize genetic risk variants. In their study, Smith et al. [23] showed that the patient-derived iPSC RPE gene expression profile is highly similar to that of their native RPE cells. Golestaneh et al. [24] generated iPSC RPE models from healthy and vAMD donors that exhibit a specific disease phenotype. Cells were stressed with different concentrations of hydrogen peroxide. AMD derived iPSC RPE cells were more susceptible to oxidative stress and had a reduced ability to upregulate super oxide dismutases (SODs) in stressful conditions. An accumulation of autophagosomes and a reduction in autophagic activity were also observed. Furthermore, iPSC derived RPE cells isolated from vAMD patients expressed higher levels of gene coding for the complement [25] and had more difficulty in attaching and surviving on nitrite-modified extra cellular matrix than cells from healthy donors. 

Finally, iPSC-RPE cells are very promising for dry AMD, which currently is incurable. Patient-derived iPSC-RPE cells could be used in autologous cell transplantation in the retina to replace damaged cells. Some clinical trials are currently underway to test these therapies.

### 3.4. Cell Cocultures and Culture Methods

#### 3.4.1. 2D Models

AMD is multi factorial disease, which involves changes in RPE cells but also in Bruch’s membrane and the underlying choroid. In order to replicate this more accurately in vitro, some studies focus on creating cell cocultures, or by cultivating cells on an artificial Bruch’s membrane.

The most commonly used biological substrates for RPE culture are collagen I and IV, fibronectin, matrigel, and gelatin. Collagen creates a layered structure with oriented fibers that mimics Bruch’s membrane and increases functionality of RPE cells. Fibronectin and matrigel are often used to coat supports, like cell inserts, to improve cell adhesion. Gelatin can be coated with different solvents (such as ethanol for example) to improve thermal stability or resistance to enzymatic degradation. These artificial membranes all aim to reproduce the effect of Bruch’s membrane. They exhibit better bio-compatibility than synthetic membranes and usually already possess cell-binding sequences, though they are sometimes not fully defined when compared to synthetic materials. Non mammalian materials, like silk fibroin and alginate have also been used [26]. These supports are permeable allowing an accumulation of macromolecular material that is shed by the cells. This could mimic the disruption of the clearance of material in vivo and accumulation of drusen [27]. 

Silk fibroin is natural silk that is obtained from the silkworm *Bombyx mori* and has traditionally been used as a suture material for centuries. It possesses several properties such as nontoxic degradation products, surface modification, and different material formats (for example sponge formats, gels, films, fibers). It can also be combined with mammalian products and is often used with a coating of collagen [28]. 

Some more ”exotic” membranes such as amniotic membranes have been used already. Amniotic membranes were obtained from human female donors by caesarean section and scraped with trypsin to remove the natural epithelium, and RPE cells were then added to the membrane with media. The amniotic membranes possess anti-inflammatory, anti-apoptotic, and epithelial cell growth promoting properties [29]. 

Synthetic materials are also used as a base layer for RPE cell culture. They are often better defined than natural materials. However, they often lack bioactivity of natural materials and have to be rendered appropriate for viable cell attachment. A commonly used material is poly(ethylene terephthalate) (PET), which is commonly coated with extra cellular matrix (ECM) proteins to enable better adherence of the cells (such as collagen, fibronectin, or laminin for example). Fetal bovine serum and poly(lactic acid) (PLA), poly(ε-caprolactone) (PCL), and poly(lactic-co-glycolic acid) (PLGA) have also been used [26]. 

#### 3.4.2. 3D Models

All these membranes are usually thin and consist of a cell monolayer. In the case of a complex diseases like AMD, replicating Bruch’s membrane involves creating a 3D membrane and, when possible, a coculture of cells. Electrospun nanofiber networks aim to replicate the complexity of Bruch’s membrane by creating a very porous material that enables the exchange of nutrients, biochemical signals, and metabolites across the membrane. Xiang et al. [30] created a structure by combining electrospun PCL (polycaprolactone, a biodegradable polyester) nanotubes with silk fibroin and gelatin. The ARPE-19 plated cells displayed a high phagocytic activity, higher polarization of pigment-epithelium derived factor (PEDF), and formation of tight junctions. Moreover, when injected in rabbits’ eyes, the electrospun membrane showed good biocompatibility and no inflammatory reactions, suggesting it would be an ideal scaffold for RPE cell transplants.

#### 3.4.3. Cocultures

Cocultures are generally considered to be more relevant to the study of AMD, as they replicate the conditions in which RPE cells are found in vivo. Most cocultures consist of choroidal or endothelial cells grown on the bottom side (basal compartment) of a culture insert (0.4 µm pore size), and RPE cells on top (apical compartment). The culture inserts are frequently coated with laminin. De Cilla et al. [31] showed that the coculture of RPE cells with HuVEC cells elicited a cross-talk between the cells when treated with aflibercept or ranibizumab, specifically through cell survival pathways and NO release. Moreover, the drugs reduced NO release in cells that had been previously treated with hydrogen peroxide. The coculture of RPE and endothelial cells (HuVEC) can in addition modulate the production of TGF-β2 and VEGFR2 expression. Indeed, RPE cells decreased the levels of VEGFR2 in HuVEC and inhibited their migration. This result suggests that angiogenic responses of endothelial cells is amplified by a decrease in TGF-β2 expression in RPE cells under pathologic conditions[32]. 

Another actor in the pathology of AMD is microglia cells. Indeed, activated microglia cells migrate to sub retinal spaces where they induce inflammation (production of NO, TNFα, IL-1β, and VEGF). Under healthy conditions, microglia cells are regulated by cytokines produced by RPE cells. In AMD, accumulated lipofuscin is phagocytosed by microglia cells and thus leads to a pro-inflammatory response and high levels of VEGF [33]. Culturing ARPE-19 cells with supernatant from activated microglia cells induced, in ARPE-19 cells, an accumulation of lipids, an increase in autophagy, and expression of pro-inflammatory genes [34]. Conversely, primary RPE cells cocultured with microglia cells activated by lipopolysaccharide produced increased levels of pro-inflammatory cytokines, presented lower levels of junctional proteins, lower levels of RPE65, and modification of cell shape [35].

Despite the accessibility and ease of use of cell culture inserts as a base for coculturing cells, they are not an accurate model for the interactions between the choroid, Bruch’s membrane, and RPE cells. A 3D model, such as electrospun fibers, is generally more relevant and should be favored when possible. Three dimensional models are more representative, both by the morphological aspects of cells and by the exposure and drug sensitivity [36]. 

A PCL and gelatin electrospun, laminin coated membrane was described by Shokoohmand et al. [37], to set up cell cocultures. Monkey choroidal endothelial cells (RF/6A) were first seeded on the bottom layer of the membrane, and after 6 days of culture, human RPE cells were seeded on the top layer. The cells were then cocultured for 20 days and multiple assays were performed. RPE cells retained their phagocytic functions, the choroid layer was found to produce more VEGF and PEDF as a coculture rather than as a monolayer of cells. This experimental procedure mimics the shift of VEGF/PEDF production in the early stages of AMD and validates the needs for cocultures as models for AMD.

Finally, a complex model of coculture replicating the choroidal stroma has been successfully engineered from a pool of human donors. It contained a base layer of choroidal stromal fibroblasts that had been grown in sheets. They were then assembled in an extra cellular matrix by placing them on top of each other and leaving them for some time to fuse together. Subsequently, either RPE, or HuVECs, or choroidal melanocytes (a type of cell found in the choroid that, in addition to providing pigmentation, is thought to reduce reactive oxygen species) were then seeded on top of the ECM. Each cell subset retained its morphological and genetic phenotype. In addition, the ECM also produced collagen and proteoglycans which can be found in in vitro choroids. Contrary to the RPE cells which formed a monolayer on top of the matrix (similar to native cells), HuVEC developed a vascular network and tubular structures. This model offers the advantage of being free of exogenous materials and of being composed only of primary cultured human cells (no transformed cell lines). It can furthermore be used to describe the interactions between multiple cells types [15]. 

### 3.5. In Vitro Stress Models

Few studies have been performed on in vitro models trying to replicate AMD, and they mostly focus on characterizing cell profiles in different conditions. However, some do focus on cell profiles either in stressful conditions, or in conditions where angiogenesis would be increased. Different methods exist to stimulate the cells in order to replicate the effects of AMD.

For example, Golestaneh et al. [24] compared iPSC-RPE derived from patients with AMD and healthy donors when treated with hydrogen peroxide, as mentioned previously. De Cilla et al. [31] also used hydrogen peroxide to stress the cells, following treatments with aflibercept and ranibizumab. They monitored the production of NOS, cell death, and PI3K and ERK1/2 activation that were found to be reduced when treated with the drugs.

To induce proliferation of cells in order to mimic angiogenesis, VEGF and FGF can be added. In an experiment by Wei et al. [14], human choroidal microvascular endothelial cells were incubated with VEGF and FGF at 20 and 30 ng/mL for 48 h. When cultured in the presence of tyrosine-kinase inhibitors targeting VEGFR and FGFR, the cells showed inhibition of proliferation, migration, and tubule formation. In a different study, human retinal microvascular endothelial cells were stimulated with VEGF and PDGF for 72 h. Similarly, proliferation was inhibited when in the presence of tyrosine-kinase inhibitors [38].

Finally, in a study to determine the effects of sorafenib (a protein kinase inhibitor for VEGFR, PDGFR, and RAF kinases), primary human RPE cells from healthy donors were incubated in PBS and illuminated under a spotlight at 300 mW/cm^2^ for an hour. They were then treated with sorafenib for 24 h. Light exposure induced cell death and the production of proliferative agents (VEGF, PGF, and PDGF), which was reduced when treated with sorafenib [39].

These models only replicate certain aspects of AMD and usually fail to regroup all the symptoms, as opposed to animal models where the disease progression more closely resembles that of human AMD. Thus, in vivo models have a high clinical significance as they are currently the best models to study AMD.

## 4. In Vivo Models of Neovascular AMD

### 4.1. Murine Models

#### 4.1.1. Induced Models

Murine models are routinely used in science as a model for complex interactions within the immune system. They are cost effective animals, with quick reproduction and disease progression, and one of the most important aspects of rodents is that transgenic models can be obtained. In the case of AMD, a laser-induced choroidal neovascularization is the most commonly implemented method in multiple species to study the disease. A targeted laser injury is performed on Bruch’s membrane and RPE cells which induces choroidal angiogenesis. First implemented in primates, the laser procedure is now used in mice and rats, despite the challenge of working with such small animals. The angiogenesis resulting from the laser treatment is similar in appearance and location to that in the human eye. This procedure is difficult to visualize in albino animals and thus only performed in pigmented rodents. Murine models are now a standard pre-requisite for vitreal injections of anti-VEGF before human trials [40]. 

Alternative methods for causing AMD-like phenotypes in mice are the injection of pro-angiogenic factors in the eye such as recombinant viral vectors overexpressing VEGF, or injection of subretinal matrigel or beads in order to cause angiogenesis. Macrophages, lipid hydroxyperoxide and polyethylene glycol have also been injected. However, none of these methods are as effective as targeted laser injury [41]. 

As previously mentioned, anti-VEGF intravitreal injection are currently the standard therapy for vAMD. However, most patients are reticent to the injections. An orally delivered treatment would be overall better received. In this way, axitinib, a multi-receptor tyrosine kinase inhibitor (that inhibits VEGFR2, PDGFRβ, and cKIT receptors, anti-angiogenic treatment currently used to treat renal cell carcinoma), has been used as a treatment in laser CNV in rat models. It was thus hypothesized that it could have a beneficial effect on vAMD. The rats were submitted to laser CVN and then treated with axitinib. As it has a short plasma half-life when dosed orally, the rats were equipped with an osmotic mini pump that sustained continued infusion. The rats treated with axitinib had reduced vascular leakage and lesions, and reduced neovascularization [38]. Oral administration to rats at a higher dose to combat its short half-life deserves to be tested as an alternative treatment in humans. 

An alternative treatment for murine CNV would be site-specific genome modification. Anti-VEGF therapy usually requires repetitive or continuous treatments over time. In the case of CRISPR, it would consist of a single long-term effect treatment by targeting *vegfa* or hypoxia inducing factor 1α (*Hif1α*). *Vegfa* gene-specific Cas9 ribonucleoproteins (RNPs) injections into the mouse eye lead to reduced choroid neovascularization [42] but a better model should target both *vegfa* and *Hif1α*. However, the usual CRISPR-Cas9 combination contains a large coding sequence (4.10 kbp) making it difficult to package with a sgRNA expression cassette into an adeno-associated virus. The Cas9 protein could have also off-target nuclease activity. Hence, LbCpf1 (from the *Lachnospiraceae*
*bacterium* ND 2006 and AsCpf1 from *Acidaminococcus* sp. BV3L6) induced DNA modifications with greater efficiency and specificity than Cas9 and Cas9 orthologs. It is also smaller than Cas9 (3.7 kpb). Delivery of adeno-associated virus-LbCpf1-Vegfa or *Hif1α* induced *vegfa* or *Hif1α* gene disruption, and a long-term reduction of the CNV area in the mouse, without cone dysfunction, at a comparable level to aflibercept. This effect was observed after a single injection, whereas aflibercept requires multiple repetitive injections [43].

#### 4.1.2. Transgenic Models

Despite multiple transgenic models, obtaining one that mimics both the early and late features of AMD is challenging, as AMD is a complex disease. It involves both genetic and environmental factors. Anatomical differences between species add to the complexity of the challenge. Though some cases of vAMD display early signs of drusen, they are usually the hallmark of dry AMD. To date, only primate models show signs of drusen that are similar in location and composition to human drusen. In the case of murine models, “drusen-like” formations can be observed in some transgenic models (such as Ccl2 knockout mice for example). However, they are comprised of an accumulation of bloated, lipofuscin-containing macrophages rather than lipids and thus it is unclear if a parallel can be made [44]. The usefulness of the Ccl2/Ccr2 knockout model is debatable, as findings differ between studies. This knockout model has been widely thought to induce drusen, thickening of Bruch’s membrane, photoreceptor malfunction, and occurrence of CNV [45]. However, Luhmann et al. [44] reported that these symptoms were also present in mice controls (WT) and were a consequence of natural ageing. Similarly, lipid bloated microglial cells were observed in a CX3CR1 knockout model. In this model, the knockout mice develop retinal degeneration and have higher sensitivity to laser induced CNV [46]. This observation suggests that this model might be more relevant than the Ccl2/Ccr2 knockout model when studying vAMD, as CNV can be easily induced. Moreover, most mice transgenic models available display dry AMD symptoms, such as RPE degeneration or, more frequently, drusen formation. It is debatable whether they could be used as vAMD models. However, they are often more sensitive to laser induced CNV, and some models display both dry and vAMD symptoms. CXCR5 is another chemokine receptor involved in AMD. CXCR5 is thought to be protective of RPE and retinal cells in ageing mice. CXCR5 knockout mice present AMD degeneration symptoms such as Bruch’s membrane thickening, amyloid-β accumulation, RPE atrophy, and spontaneous neovascularization and drusen [47,48].

In humans, genetic studies have shown a link between cholesterol-related genes and AMD, such as ApoE for example (a glycoprotein responsible for the distribution of cholesterol and lipids among cells). ApoE murine models have been created which express one of the three human ApoE isoforms under the control of mouse *apoE* regulatory sequences. With ageing and a high-fat diet, mice expressing the ApoE4 isoform expressed multiple hallmarks of human AMD: drusen-like deposits, a thickening a Bruch’s membrane, retinal and choroid vascularization, and RPE degeneration. ApoE4 seems to accelerate amyloid β accumulation [27]. Moreover, lipid accumulation leading to the thickening of Bruch’s membrane has been linked to uptake and clearance by RPE cells through CD36. Oxidized LDL are CD36 dependent, and CD36−/− mice resulted in increased lipid sub-retinal deposits. Expression of CD36 preserved visual function by reducing deposits [49].

Oxidative stress models are also frequently used as they display multiple symptoms of AMD. The retina is highly susceptible to oxidative stress as it is exposed to light and high levels of oxygen and polyunsaturated fatty acids. SOD is an antioxidant system that catalyses superoxide radical dismutation. It is comprised of three isoenzymes, of which SOD1 has the highest activity. Mice in which *Sod1* was knocked out, develop drusen, and even more so when exposed to light. They also developed RPE degeneration and dysfunction, and a small portion of mice developed CNV and vascular leakage compared to none in the control groups. The symptoms were exacerbated with age [2]. The nuclear factor erythroid 2-related factor 2 (NRF2) also plays a key role in regulation and prevention of the oxidative stress. *Nrf2^−/−^* mice showed age-dependent development of CNV, degeneration of RPEs and of chorio-capillaries, and drusen-like deposits compared to wild-type controls [50]. 

Double CXCR5/NRF2 knockout mice developed AMD symptoms in younger animals (4–6 months old), thus reducing the waiting time for retinal degeneration. The double knockout mice presented sub-retinal deposits, enlarged retinal vessels, receptor apoptosis, and increased microglial markers (TMEM119) [51].

Several other knockout or transgenic models exist, for example in the complement pathway (complement factor H knockout, C3 overexpressing mice to name a few). These models all exhibit some features of AMD, but rarely an accumulation of multiple symptoms and are often associated with dry AMD as they induce the formation of drusen and retinal degeneration rather than CNV [41]. However, transgenic models hold promising potential for the study of vAMD, as they are chronic models rather than acute injuries (like laser induced CNV for example) and thus could more accurately transcribe the complex mechanisms that eventually lead to AMD. In order to most accurately mimic the human pathology, it is necessary to use transgenic animals and then potentially induce the CNV by laser or injection.

A mutation of the *Crb1 (Rd8)* gene is prevalent in multiple mice strains, notably the most widely used C57BL/6N strain. This autosomal recessive single nucleotide deletion in the *Crb1* gene, results in retinal degeneration. This mutation was found in all commercial sources of C57BL/6N, but not in the C57BL/6J sub-strain. Affected mice display ocular lesions. This phenotype could lead to false interpretations of the effect associated with a specific transgene or a specific knock-out. It is therefore important to screen the mice for *R8* prior to any study and backcross them if necessary [52].

### 4.2. Zebrafish Models

#### 4.2.1. Induced Models

Though mouse models provide great insights into the progression and treatment of the disease, there is more and more pressure to step away from experimentation in mammals. The zebrafish has gained in popularity, as it is a good model for comparison between vertebrates, as well as for developmental, genetic, behavioral and environmental studies [53].

Zebrafish have the benefit of being a good recipient for treatments, as the drugs of choice can be added to the culture medium rather than injected in the fish. They also have lower upkeep costs and require less resources (using Petri dishes to grow the fish for example) and have high reproduction rates. Zebrafishes have the added benefit of having transparent bodies, which allows points of interest to be closely monitored.

Around 70% of genes in the human genome have orthologs in the zebrafish genome. In some instances, zebrafishes have two orthologs for a single gene in humans (for example *vegfa* orthologs: *vegfaa* and *vegfab*). This leads to potentially different expression profiles that can be used in knockout models. With the advent of CRISPR, cells can be marked for imaging, transgenic constructions generated, and expressing or inhibiting genes of interest. In addition, the visual organs of adult zebrafish revealed that it is highly comparable to that of mice and humans including an anterior cornea (analogous to eyelids in mammals), posterior cornea, spherical lens (as in mice), thin vitreous retina, porous RPE, Bruch’s membrane, chorio-capillaries, and outer choroidal structures including a rete mirabile (analogous to the choroid body in humans and mice) and sclera [54] with a choroid, RPE cells, and photoreceptors. They are diurnal and possess the ability to see red, green, and blue wavelengths (in addition to UV possessing receptors) [55]. 

As such, zebrafish can serve as an early model for testing of anti-VGFR drugs, both by determining the cytotoxicity of the drug and its effect on angiogenesis. 

Lenvatinib is a multi-targeted tyrosine kinase inhibitor against VEGFR1/2/3 kinase, as well as fibroblast growth factor receptors (FGFR), PDGFR, cKIT receptor, and the RET proto-oncogene. Wei et al. [14] showed that lenvatinib suppressed angiogenesis of subintestinal vessels in zebrafish. They used choroidal cells (human choroidal microvascular endothelial cells) as a model in vitro, and zebrafish in vivo, where they looked at subintestinal vessel formation in transgenic embryos (endothelial cells expressing mCherry). They then confirmed the efficacy of lenvatinib on vAMD in a mouse model. 

Similarly, Li et al. [56] showed the same effects in embryonic zebrafish angiogenesis using brivanib, an inhibitor of FGFR and VEGFR. They also used embryos expressing mCherry under the control of the VEGFR2 promotor (endothelial cells expressing mCherry). They labeled motor neurons with GFP in order to monitor potential neuronal damage, and subsequently did not see any detrimental effects of brivanib on neural development. Despite these models showing a reduction of embryonic angiogenesis, their studies were not directed towards disease or stress induced angiogenesis. 

Finally, Cao et al. [57] showed that a reliable model of neovascularization could be induced in fli:egfp zebrafish (endothelial and lymphatic vessel expressing GFP) by putting them in hypoxia. Hypoxia-induced transcription factor (HIF, composed of two subunits, HIF-α and HIF-β) is a regulator of hypoxia-inducible genes and genes associated with angiogenesis. Under normal circumstances, HIF-α is hydroxylated by a protein complex that possesses an ubiquitin ligase E3 activity, encoded by von Hippel–Lindau (VHL) and targeted for proteosomal degradation. In hypoxia, VHL is inactivated inducing the stabilization of HIF-1α. HIF-α and HIF-β are translocated to the nucleus and form a functional HIF protein, which induces the transcription of multiple angiogenic growth factors, such as VEGF, PDGF, and CXCR4. VEGFA acts as the key regulator of angiogenesis as it facilitates blood vessel growth and increases vascular permeability. Overexpression of hypoxia inducible mRNA (like VEGFA) is the hallmark of exacerbated vascularization, notably encountered in vAMD. At 10% O_2_ for 15 days, the zebrafish exhibited significantly more retinal neovascularization than in control groups. Furthermore, by adding VEGFR inhibitors (sunitinib) to the water containing the zebrafish, inhibition of angiogenesis and normalization of the retinal vascularization were observed. This induced vascularization makes the hypoxia model relevant for the study of vAMD in zebrafishes.

#### 4.2.2. Transgenic Models

A zebrafish model in which the RPE cells are genetically removed is available. A promoter element drives the expression of an *E. coli* nitroreductase, which converts metronidazole (an antibiotic and antiparasitic used for the treatment of protozoa and anaerobic bacteria) into a DNA crosslinking agent and leads to the apoptosis of expressing cells. The RPE cells and Bruch’s membrane both degenerate and after a few months they regenerate and function as normal. These results were observed both in embryos and in adult fishes. The authors hypothesize that the cell regeneration could be driven by the Wnt signalling pathway. They suggest that with more research and better understanding of these mechanisms this method could be used for the treatment of dry AMD, as it is related to the degeneration of RPE cells [58]. Some zebrafish mutants like gantenbein (gnn) and pde6c have a genetic mutation that causes the loss of cones and that later develop pathologies to RPE cells. The gnn zebrafish for example develops dystrophic red cones that then evolve in a full degeneration of cones and RPE cells [59]. In the case of pde6c mutants, the cone death is initiated through a RIP3 kinase (receptor interacting protein 3)-dependent pathway, which is a regulator of necroptotic cell death. When the *rip3* gene was knocked out, the visual response of pde6c larvae was restored [60]. It is yet to determine if these transgenic models could be used for vAMD, as they do not seem to develop neovascularization. In order to simulate vAMD, genetic models should induce angiogenesis when possible, or coupling transgenic models with other modifications promoting vascular leakage in the eyes. 

In humans, inactivation of the *von Hippel–Lindau* (*VHL*) tumor suppressor gene predisposes humans to develop highly vascularized neoplasms. Van Rooijen et al. [61] showed that in VHL knockout mutant zebrafish the phenomena was reproduced. In the absence of VHL, HIF-1α is stabilized (which mimics hypoxia) and induces the transcription of angiogenic factors. Vhl-/- zebrafishes develop severe neovascularization in the brain, eye, and trunk. They also observed severe vascular leakage, edemas, and retinal detachment. There was an increase in VEGFR1/2 signalling in embryos at up to 7 days post fertilization. When treated with two multi-targeted VEGFR tyrosine kinase inhibitors (676475 (Calbiochem) and sunitinib), a complete inhibition of all angiogenesis abnormalities, notably the eye vasculature was observed. Hence, VHL knockout zebrafishes could represent a good model for vAMD. Complementarily, *HIF-1α* knockout fishes display severe impairment of blood vessel formation. HIF-1α is required for developmental angiogenesis, modulation of interactions between macrophages and endothelial cells and vessel repair after hypoxic conditions. These observations further support the idea that abnormal or overexpression of HIF could lead to uncontrolled angiogenesis [62].

Finally, as previously mentioned, the supplementation of diets with vitamins delays the onset of AMD in humans. Injection of zeaxanthin, a carotenoid that occurs naturally in the eye, into the zebrarish’s eye increases visual acuity compared to the control groups. The authors made a parallel with the studies performed in humans and hypothesized that intraocular injections of zeaxanthin instead of oral supplementation could have beneficial effects in human AMD patients [63]. However, the limitations of this study are that fishes without AMD-like symptoms were used.

## 5. Conclusions

AMD is a widespread disease that affects many people worldwide. vAMD is currently treated with intravitreal injections of anti-VEGF. However, this method is lacking. Though it improves the loss of vision by reducing angiogenesis and sometimes stopping it, it is time consuming and unpleasant for the patients. Many alternatives are being tested, namely anti-VEGFs that could be taken orally. With the aim of better understanding the disease and its mechanisms, many in vitro and in vivo models have been devised (Table 1).

In vitro, ARPE-19 cells and stem-cell derived iPSC-RPE cells are often used as models to understand cellular interactions by being cultivated on 2D and 3D model membranes. Cocultures with choroid or epithelial cells enable us to better understand communication between cell types. By stressing the cells with hydrogen peroxide or high intensity lighting, or stimulating their proliferation with VEGF and FGF, novel treatments can be tested. Anti-VEGF drugs are the most promising when treating vAMD and will hopefully be delivered by other means than intravitreal injections. However, in order to properly assess the complex interactions between the patient and the disease, in vivo models are the closest and most reliable models available besides humans.

Murine models are particularly interesting as a form of AMD that can be induced by laser-induced choroidal neovascularization. They show promising responses to anti-VEGF agents, and have shown that the use of site-specific genome modification could potentially be considered as a therapy. Zebrafishes are emerging as a relevant model for AMD due to their similarities with the human eye and genome. They are transparent and it is thus very easy to observe the effects or toxicity of anti-angiogenic drugs on them. Finally, AMD is a complex disease that requires more research through the use of in vitro and in vivo models.

## Figures and Tables

**Figure 1 ijms-21-04627-f001:**
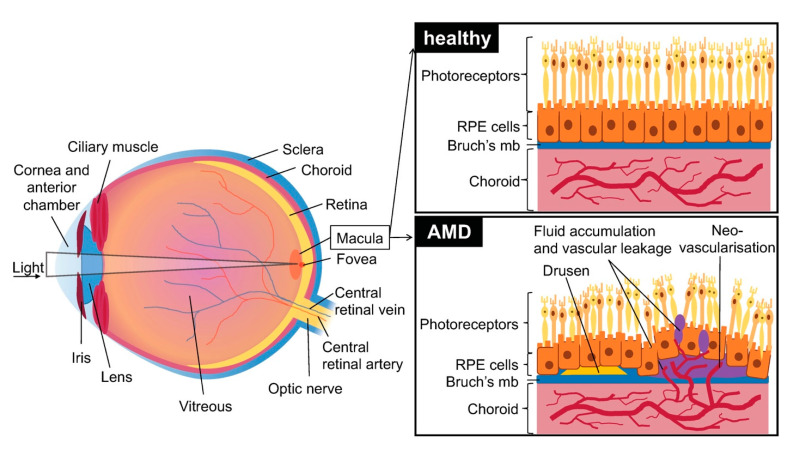
Schematic of the eye with healthy retina, choroid and retinal pigment epithelium (RPE) cells and effects of age related macular degeneration (AMD): presence of early drusen and neovascularization, disruption of Bruch’s membrane and RPE cells, and vascular leakage. Brusch’s mb = Brusch’s membrane.

**Table 1 ijms-21-04627-t001:** Advantages and disadvantages of neovascular models.

Models	Advantages	Disadvantages
In Vitro		
**Cell Lines**	Reduced donor-to-donor variability. Defined models with structured experimental conditions and good reproducibility.	A single cell type will not reproduce systemic defects. Does not reproduce the complexity of interactions in a living model. Cells grow unnaturally fast and gene and protein expression are often vastly different than in vivo.
PrimaryRPE cells	Human cells with natural differentiation. Mimic perfectly human pathology.	Can be used for only a few passages. Need to have access to this type of sample and the related authorizations
ARPE-19 cells	Immortalized, rapid cell growth. Exhibits similar morphology and genetic makeup to primary RPE cells.	Incomplete polarization of the cells compared to primary RPEs.
IPSC-RPE	Exhibits similar morphology and genetic makeup to primary RPE cells. Patient derived iPSC-RPE cells could be used in autologous cell replacement therapy.	Differentiation of the cells is time consuming and requires growth factors.
**Cocultures**	Enables cell-to-cell interactions and cross-talks, and modulation of cytokine production. More cytokines are usually produced in cocultures.	
2D models	Easier to put in place than 3D models, better for long-term cultures.	Lack of sophistication, cells grow in monolayers at the same speed. Drugs are up-taken more easily than they would in vivo which is less accurate.
3D models	Multi layered, the cells can self-organize. The most accurate in vitro representation of the choroid, Bruch’s membrane and RPE cells. Cells develop vascular networks and can migrate. More representative of drug exposure. Gene and protein levels closer to those in vivo. Better cell junctions.	More resource intensive, electrospun scaffolds require specialized equipment, in addition to more time and expertise. Can be difficult to replicate.
**In Vivo**		
**Murine Models**	Most retinal degeneration genes in mice have a corresponding gene in humans, many human gene orthologs in the mice genome. They have short lifespans which enables us to see the ageing process. Protocols for genetic studies are well established.	Mice do not possess a macula. They do not produce drusen that are similar in location and composition to human drusen. Pathogenesis can differ. Late onset genetic models can lead to waiting over a year.
Laser induced CNV	Replicates the neovascularization in neovascular AMD, is low cost, and the CNV develops rapidly.	It is an acute injury rather than a chronic one, and thus has the inability to reproduce the complex events that lead to AMD. Risk of cataract and fibrosis if the procedure is not performed correctly.
Injection induced CNV	Simulates the exudative deposits and lesions in neovascular AMD.	Lower incidence of CNV than laser induced. Injections can cause tears in Bruch’s membrane.
Injection of adenovirus	Injection of vectors expressing VEGF have high incidence and long-term capability to induce CNV. These models work well on transgenic models.	
Transgenic models	Increased sensitivity to laser induced CNV, some models develop CNV with age. They are more complex models than acute induced CNV.	Most transgenic models exhibit dry AMD symptoms, such as drusen formation and retinal degeneration, and therefore will need injection or laser induced CNV to simulate neovascular AMD, making the experiment more costly and time consuming.
**Zebrafish Models**	Common features in retinal vasculature, many human orthologs in the zebrafish genome. Cost effective. Accessibility of screening and study of vascular patterning. Easy to treat.	Fewer models that replicate AMD, as zebrafish are less used than murine models. Pathogenesis can differ.
Hypoxia induced	Non-invasive induction of angiogenesis, easily reproducible and low cost. Can be induced in transgenic fish.	
Transgenic models	VHL knockout models exhibit high vascularization. These models are easy to work with as treatments are easy to deliver.	

RPE = retinal pigment epithelium; AMD = age related macular degeneration; CNV = choroidal neovascularization.

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
