# Peer review of "Experimental Models in Neovascular Age Related Macular Degeneration"

_ijms, 2020, doi:10.3390/ijms21134627_

Round 1

Reviewer 1 Report

Manuscript “Experimental models in neovascular Age related Macular Degeneration” Rastoin et al. is a well-conducted and organized review of the recent developments in the field of neovascular AMD. The manuscript is well written and presented and overall suitable for publication in IJMS. Please see a few minor points and suggestions below.

Minor points:

  1. Very representative graphic summary in Figure 1, authors need to clarify the source of the image, was it done by the authors themselves or outsourced to the scientific artist. If the latter, the copyright transfer is required. If the authors used any scientific art online resource, it is also should be mentioned. 
  2. ARPE 19 cells – authors need to mention that ARPE-19 cells do not fully recapitulate the transcriptome of primary RPE cells. Please refer to the study Samuel et al. (Mol Vis. 2017; 23: 60–89. http://www.molvis.org/molvis/v23/60) who report only partial transcriptome consistency between ARPE-19 and primary human RPE cells even after an extended period of in vitro differentiation.
  3. Authors need to describe human primary RPE cultures – that are fairly easy to isolate and proliferate up to 7-10 passages. Human cadaverous eye cups discard from cornea transplantation needs is a good source of primary human RPE cells, with fetal eyes from the abortion material is another. Authors need to add more perspective of the pros and cons of primary human material and cell lines and IPS derived RPE cells.
  4. 2D models – Authors can mention good proliferation and differentiation of primary RPE cells in cellulose support as reported by Johnson et al. (10.1073/pnas.1109703108) and cellulose support to be useful to study the extracellular deposition of the RPE cells. 
  5. Cell co-cultures and culture methods – Authors put a lot of focus on the role of RPE cells in nAMD, however, it would be more complete to discuss the microglia effects in the context of retinal degeneration and  nAMD.
  6. Animal models - Authors should address the CRB1-RD8 mutation issue in the transgenic animal lines that affect AMD research in the field. The topic described well in Invest Ophthalmol Vis Sci . 2012 May 17;53(6):2921-7. doi: 10.1167/iovs.12-9662.
  7. Authors discuss SOD1 mice, but do not mention NRF2-/- mice such as (Zhao et al. 10.1371/journal.pone.0019456).
  8. The authors can also mention the rapid degeneration model of CXCR5/NRF2 (Huang et al. 10.1016/j.exer.2020.108061).
  9. Similarly, we have recently presented CXCR5-deficient animals as the model of retinal degeneration with AMD associated proteins accumulation and nAMD component. (Huang et al. 10.1371/journal.pone.0173716) and (Lennikov et al. 10.3389/fimmu.2019.01903)
  10. Zebrafish models – Authors mention Dr. Lasse Jensen's hypoxia-induced neovascularization in the zebrafish model. However, they should also discuss the differences in zebrafish neovascularisation compared with the mammalian system, this is well described in (Ali et al. 10.1161/ATVBAHA).
  11. Conclusion: If authors have an opinion regarding the most promising direction on nAMD research from their summary they should state it in the conclusion.

Author Response

We thank the reviewer for for these very positive comments and his particularly relevant remarks and suggestions. The graphic was made by ourselves. We have implemented our manuscript with these suggestions (marked in red in the revised manuscript).

Reviewer 2 Report

The article makes an extensive review of the different experimental models, both in vitro and in vivo, for the study of neovascularization and provides a very practical view of the different models as well as a highly valuable summary table for those who want to use any of these models. The article is very well written, with a lot of criteria and a clear order that makes it very easy to read and obtain useful information.

I recomend to the author to review in the introduction, when discussing current approved therapies, that Bevacizumab is not approved for ophthalmic use in all countries and that it is used only off label in many of them. I recomend also to review certain extra spaces between words.

Author Response

We thank the reviwer for these very positive comments. We have corrected and specified that bevacizumab is not officially approved for the treatment of AMD, as suggested by the reviewer.